

# Study on the induction of exogenous plant hormones to enhance the weed suppression ability of allelopathic and non-allelopathic rice accessions

Ting Wang[1,*], Xinyi Ye[2,*], Yuhui Fan[1], Shuyu Chen[2], Huayan Ma[2] and Jiayu Li[2]

[1] College of JunCao Science and Ecology, Fujian Agriculture and Forestry University, Fuzhou, Fujian, China
[2] College of Life Sciences, Fujian Agriculture and Forestry University, Fuzhou, Fujian, China
[*] These authors contributed equally to this work.

## ABSTRACT

**Background.** Increasing the induced-allelopathic potential of rice at the seedling stage by chemical induction is an important strategy in weed management. More in-depth work is needed to find inducing agents for effectively improving the weed-suppressive activity of allelopathic and non-allelopathic rice accessions *via* the modulation of root morphology and allelochemicals production.

**Methods.** Grown in Hoagland's solution, two rice cultivars—allelopathic PI312777 (PI) and non-allelopathic Lemont (LE)—were treated using various phytohormones to evaluate root growth and allelopathic potential. Optimal phytohormones (ABA and EBL) were selected based on root growth responses. Extraction of rice leaf and root exudates, subsequently assessed on lettuce seedlings, revealed induced allelopathy. WinRHIZO software was used to investigate root morphologies; HPLC and GC-MS evaluated phenolic acids and terpenoids in rice tissues. The transcription of genes related to phenolic acid and terpenoid biosynthesis was measured by qPCR. ANOVA and correlation analysis were applied in data analysis.

**Results.** The results showed that the optimal induction conditions were abscisic acid (ABA, three $\mu$mol/L) or 24-epibrassinolide (EBL, 0.5 nmol/L) for 3 days. Under these conditions, the induced-allelopathy (IA) values of rice root exudates induced by three $\mu$mol/L ABA were 9.62% in PI31277, and 13.76% in Lemont, while the IA values of PI31277 and Lemont after 0.5 nmol/L EBL treatment were 7.83% and 11.51%, respectively. ABA mainly affected the root length and volume at 0–0.2 mm diameter, while EBL mostly affected root length and volume at 0–1.0 mm diameter. The total content of phenolic acids in root-exudates of PI31277 and Lemont induced by EBL were 1.16 and 2.33 times greater, respectively, while ABA induction also increased the phenolic acid content by 1.48 and 1.87 times, respectively. Most genes related to phenolic acid and terpenoid synthesis, such as *PAL*, *C4H*, *F5H*, *MK*, *SQS*, and *PS*, were upregulated after EBL induction, while most genes were downregulated after ABA treatment. Moreover, correlation analysis confirmed that root morphology, allelochemicals production, and related gene expression levels were significantly correlated with induced-allelopathy caused by ABA or EBL treatment.

Corresponding author
Jiayu Li, lijiayubnu@126.com

## INTRODUCTION

Weed hazard is a global challenge to agricultural production. Various herbicide reduction or replacement technologies in farmland have been developed and attempted (*Macias, Mejias & Molinillo, 2019*). In particular, rice allelopathy-based weed control shows promising progress (*Chung et al., 2018*). Rice's allelopathic effects emerge at 3–6 leaf stage, and weeds in the first 30 post-transplant days minimally affect yield. Environmental factors enhance rice allelopathy but are impractical for field application (*Wang et al., 2010*; *Kato-Noguchi & Peters, 2013*; *Khanh et al., 2018*). Studies have shown that rice allelopathy is a chemically induced defense response mechanism, in which rice can recognize root exudates of associated weeds and initiate response mechanisms to release more allelochemicals and improve weed-inhibition (*Bi et al., 2007*; *Fang et al., 2009*). The use of exogenous compounds such as salicylic acid, methyl jasmonate, and methyl salicylate can improve rice allelopathy (*Xu et al., 2010*; *Kong et al., 2004a*; *Kong et al., 2018*; *Guo et al., 2017*). *Zhang et al. (2018)* also demonstrated that the allelopathic potential of allelopathic rice and non allelopathic rice can be induced by barnyardgrass root exudates. The genetic allelopathy (GA) exhibited by the allelopathic rice variety PI312777 is markedly greater than that of the non-allelopathic rice variety Lemont. However, the induced allelopathy (IA) levels were approximately comparable between the two accessions, with the IA of the non-allelopathic rice Lemont being even more pronounced. This result highlights the importance of induced allelopathy for the first time (*Zhang et al., 2018*). Since GA naturally occurs in only 3–5% of rice accessions and is often lost during breeding (*Courtois & Olofsdotter, 1998*). IA provides a practical alternative for enhancing weed suppression in non-allelopathic varieties. These results remind us that increasing the induced-allelopathic potential of rice at the seedling stage by chemical induction may be a new approach for the application of allelopathy in rice production.

Rice suppresses weeds mainly *via* root-secreted allelochemicals. Phenolic acids, terpenoids, and flavonoids are commonly studied allelochemicals in rice (*Kong et al., 2004b*; *Kato-Noguchi, 2011*; *Fang et al., 2016*; *Li et al., 2020*). The allelopathic rice could synthesize phenolic acids through the phenylalanine metabolic pathway to inhibit the allelopathic effect of adjacent barnyardgrass. Phenylalanine ammonia-lyase (PAL) is the rate-limiting enzyme in the phenylalanine metabolic pathway, and its activity is crucial for regulating phenolic acid biosynthesis in rice tissues. Research by *Bi et al. (2007)* indicates that the external application of methyl jasmonate and methyl salicylate can enhance the activity of PAL enzyme, and facilitated the accumulation of phenolic acids in rice. *Zhang et al. (2019)* demonstrated that the expression levels of *PAL*, 4-coumarate-CoA ligase (*4CL*), and cinnamoyl-CoA reductase (*CCR*) in the leaves of both allelopathic and non-allelopathic rice were upregulated after induction with barnyardgrass root exudates. Moreover, it enhanced the total secretion of phenolic acids by rice roots, thereby improving the

allelopathic induction in rice. Another important class of rice allelochemicals is terpenoids, especially momilactone A and B. In addition, *Li et al. (2020)* found that the total content of terpenoids and monoterpenes in rice with allelopathic effects and the surrounding soil was significantly higher than that in non-allelopathic rice. Additionally, the proportion of oxygenated monoterpenes in the outer soil of allelopathic rice was greater. Some exogenous biotic and abiotic stimuli, including fungi, jasmonic acid, cold stress, *etc.*, can induce the transcription level of momilactone substance synthesis genes in rice tissues (*Hasegawa et al., 2010*; *Yokotani et al., 2013*; *Yoshida et al., 2017*). Notably, chemical induction enhances rice allelopathy by upregulating genes in allelochemical biosynthesis pathways and increasing allelochemical production.

The allelopathy between rice and weeds relies on the involvement of the root system in the underground part of the plant. Activated carbon adsorption, root segregation, and allelochemical profiling distinguish allelopathy from resource competition (*Yang & Kong, 2017*; *Fernandez et al., 2016*). The allelopathic effect of rice plays an important role in the continuous interference of weeds in rice fields at the root level, which can adjust the distribution pattern of roots based on the presence of rice with allelopathic characteristics (*Yang & Kong, 2017*). When coexisting with its kin, allelopathic rice could enhance its inhibitory effect on nearby barnyardgrass by regulating its root behavior pattern, allelochemical contents and rhizosphere microbial community in the soil (*Xu et al., 2021*). *Li et al. (2019)* found that rice with allelopathic effects, when cultivated in hydroponic culture had a greater number of root tips and increased root biomass, primarily consisting of finer roots. Moreover, in 0–5 cm vertical and 6–12 cm horizontal soil layers, allelopathic rice exhibited greater root densities and higher benzoic acid derivatives than non-allelopathic rice. Based on these research advances, we hypothesized that rice root morphology at the seedling stage could be used as the induction target, and exogenous chemicals could regulate root growth and distribution to increase the content of allelochemicals secreted by rice roots, which would further enhance its weed-suppressive potential at the seedling stage. In this study, we investigated the induction of allelopathic properties in common exogenous hormones in rice by using root morphology at seedling stage as a screening target in hydroponic experiments. We evaluated the changes of induced-allelopathy of rice with different allelopathic potentials under the exogenous application of different plant hormones. In addition, we further explored the underlying inductive mechanism by analyzing root morphology, allelochemicals production, and the expression levels of allelochemical synthesis related genes.

There is a wealth of information that some classical plant hormones such as abscisic acid, gibberellin, and indoleacetic acid, *etc.*, or new hormones such as salicylic acid, jasmonate, methyl jasmonate, brassinolide, and strigolactones, *etc* can regulate the growth and development of plant roots individually or interactively (*Zhu et al., 2006*; *Chen et al., 2006*; *Haider et al., 2018*; *Zhou et al., 2019*). 24-Epibrassinolide enhances the antioxidant enzyme activity in rice plants and promotes root system development (*Divi & Krishna, 2009*; *Chen et al., 2023*). Abscisic acid bidirectionally regulates root growth by modulating auxin signaling pathways and reactive oxygen species (ROS) metabolism. Furthermore, low concentrations of ABA stimulate root elongation and enhance the accumulation

of phenolic compounds (*Harris, 2015*; *Xu et al., 2013*). Nevertheless, there are limited accounts regarding the utilization of exogenous plant hormones to improve rice grass-resistance by regulating rice roots at the seedling stage and even fewer applications. In summary, these efforts can generate fresh concepts and scientific foundation for the sustainable advancement of agriculture by reducing the use of chemical herbicides.

## MATERIALS & METHODS

### Rice seedling cultivation and induction treatments

An allelopathic rice accession PI312777 and a non-allelopathic rice accession Lemont, which had been identified previously (*Dilday, Lin & Yan, 1994*) and obtained in our lab, were used as target plants for induction. The induction experiment with PI and LE was conducted following the manner detailed in our prior paper (*Zhang et al., 2018*). Briefly, the homogeneous rice seedlings of each accession at the 1-leaf stage were cultivated in a plastic cup containing 0.4 L of Hoagland's solution, with one rice seedling per cup. At the four-leaf stage, the culture solution was switched new Hoagland's solution (pH 6.0) containing different induction treatments as subsequent experiments. Hoagland's solution without an inducing solution was used as a control. The experiments followed a completely randomized design, with twenty repetitions of each treatment. Rice was grown in a growing space with diurnal and nocturnal temperatures of 20−30 °C and relative humidity varying from 65% to 90%.

### Selection of appropriate plant hormones with optimum induction conditions

Experiment 1 was run to evaluate the changes of the induced-allelopathy of PI and LE treated with the optimum induction phytohormones. In the preliminary experiments, a total of 10 plant hormones with three concentrations were used to investigate their effects on the root growth of two rice cultivars ('PI' and 'LE') in hydroponic solution. Based on the preliminary screening results (see results in Table S1), four exogenous plant hormones were selected for the induction dose experiments, including ethephon (ETH) (0, 0.17, 0.34, 0.68, 1.36 μmol/L), abscisic acid (ABA) (0, 1, 3, 5, 7 μmol/L), brassinolide (BR) (0, 0.01, 0.03, 0.05, 0.07 nmol/L), and 24-epibrassinolide (EBL) (0, 0.1, 0.3, 0.5, 0.7 nmol/L). All compounds were first dissolved in a small volume of ethanol to prepare a high-concentration solution according to the needs of the experiment, and then diluted with Hoagland's solution into the above different final concentrations which were used for rice cultivation. Seven days post-treatment, the rice leaves were harvested, sectioned into two cm segments, and immersed in distilled water (8% w/v) for 24 h in an incubator set at 25 °C. The extracts were filtered with a sterile 0.22 μm membrane filter for the subsequent bioassay, which can estimate the induced- allelopathy of two rice cultivars.

Experiment 2 for the effect of treatment duration was similar to the first experiment described above. The dosages were set as ABA (three and five μmol/L) and EBL (0.3 and 0.5 nmol/L), which was based on the results of Experiment 1 above (see results in Figs. S1, S2). The treatment duration was set at 1, 3, 5, and 7 days. The preparation of the rice leaf extracts and their bioassays were conducted as previously described in Experiment 1.

The results of two experiments above determined which induction dosage (three μmol/L ABA, 0.5 nmol/L EBL) and treatment duration (3 days) of two appropriate phytohormones (ABA or EBL) were selected for further validation. Hoagland's solution without inducing solution was used as control. The preparation of the rice leaf extracts for the subsequent bioassay were conducted as previously described in Experiment 1. In addition, rotary evaporation at 40 °C ± 1 °C was used to collect and concentrate the culture solution of each treatment to 200 mL. The concentrated culture solutions were refrigerated at 4 °C for 24 h, then filtered through a 0.22 μm filter and diluted to a volume of 200 mL, which served as the concentrated root fluids for subsequent laboratory bioassays.

### Bioassay of the induced-allelopathy of two rice cultivars

The alteration in rice-induced allelopathy was assessed by quantifying the inhibiting effects of rice extracts from leaves or rice root exudates on the root length of lettuce (*Lactuca sativa* L.) in a laboratory bioassay, as previously detailed (*Zhang et al., 2019*). Briefly, five mL of the rice leaf extracts or concentrated root exudates obtained from the induction experiments above were introduced into tissue culture flasks that were lined with filter paper at the bottom. Hoagland's solution was serving as a control in this experiment. Five sprouted lettuce seedlings were sowed on filter paper and subsequently positioned in a Petri dish, which was subsequently set in an incubator sustained at 25 ± 2 °C, with a daily light cycle of 12 h from 6:00 to 18:00. Each experimental trial was conducted five times, and the root length of the lettuce was assessed within three days.

The inhibiting ratio (%) was employed to evaluate the impact of the experimental solutions compared to the control, calculated using the following formula: inhibition rate (%) = (1−treatment/control) ×100%. The inhibition rate of rice accessions on lettuce with and without induction was defined as total allelopathy (TA) and genetic allelopathy (GA), respectively, and the increment of inhibition rate was defined as induced-allelopathy (IA), wherein IA = TA–GA (*Zhang et al., 2018*).

### Fine-root trait calculation of two rice cultivars

To investigate the impact of exogenous plant-produced hormones on the root morphology of Rice PI with allelopathic effects and rice LE without allelopathic effects, the optimum induction conditions (that is, three μmol/L ABA or 0.5 nmol/L EBL, and 3 days of treatment) were used in a hydroponic experiment. Hoagland's solution without inducing solution was used as a control. After the seedlings were harvested, the pristine and undamaged roots were scanned using an Epson Expression LA2400 scanner (Seiko Epson Co., Nagano-ken, Japan) to produce a grayscale image. The image was analyzed using WinRHIZO software (Regent Instruments Inc., Quebec, Canada) to extract morphological characteristics following the methodology of *Li et al. (2019)*. Rice roots were investigated in three diameter ranges (0–0.2 mm, 0.2–1.0 mm, and 1.0–2.0 mm) for the evaluation of three fine-root morphology traits (root length, root surface area, and root volume). The number of root tips for the entire root was also obtained. The root biomass was obtained by oven-drying the roots at 105 °C for 30 min and 80 °C for 48 h.

## Analysis of phenolic acids and terpenoid contents

The quantification of phenolic acids in rice leaves and root exudates was conducted utilizing a combination of liquid extraction and solid-phase extraction, subsequently analyzed through high-performance liquid chromatography (SPE-HPLC). After three days of treatment with three μmol/L ABA or 0.5 nmol/L EBL, the leaf extracts or concentrated root exudates were extractedusing Cleanert PEP solid phase extraction cartridges (Agela Technologies, Tianjin, China). Methanol was added after the cartridge was diluted with water, and then concentrated with N2 before being diluted in 500 μL of methanol for quantitative analysis by HPLC. The quantitative analysis of seven phenolic allelochemicals, namely, protocatechuic acid, 4-hydroxybenzoic acid, syringic acid, vanillic acid, salicylic acid, ferulic acid, and cinnamic acid, was undertaken as previously outlined (*Li et al., 2022*). The phenolic acid contents in rice leaves were quantified as the number of micrograms per gram of fresh leaves. The concentration of phenolic acids in the root exudates was quantified in micrograms per liter of the rice culture solution.

After three days of treatment with three μmol/L ABA or 0.5 nmol/L EBL, the rice leaf extracts were sampled. The contents of terpenoids in rice leaves were determined by microwave-assisted extraction and semi-quantitative analysis by GC-MS, depending on the method reported by *Li et al. (2020)*. The concentrations of terpenoids were determined by multiplying the area of the measured terpenoid peaks by the quantity of the internal standard (n-dodecane), and then this product was divided by the product of the measured peak area of the internal standard and the weight of the soil sample.

## Quantitative real-time PCR

Based on the RNA-seq results of *Zhang et al. (2019)*, we selected ten key genes involved in the phenolic acid (*PAL*, *C4H*, *F5H* and *COMT*) and terpenoid (*HMGR*, *MK*, *MTS*, *STS*, *PS* and *SQS*) biosynthesis pathways for quantitative real-time PCR (qPCR) analysis. Rice roots and the second internode leaf were sampled at 3 d after treatment with three μmol/L ABA or 0.5 nmol/L EBL as described above. The relative expression of each target gene was assessed using the $2^{-\Delta\Delta Ct}$ method (*Zhang et al., 2018*) and a bar chart was drawn representing the expression level values of 10 key genes induced by ABA or EBL compared with the control of the same rice cultivar. All primers required to perform qRT-PCR are given in Table S2.

## Statistical analysis

Prior to statistical analysis, homogeneity of variance was confirmed by residual plots, and normality was verified using Q–Q plots and Shapiro–Wilk's normality test. The analysis of the significance of differences among various treatments applied to the same rice cultivars was conducted using one-way ANOVA utilizing SPSS (Version 20; SPSS Inc., Chicago, IL, USA), followed by Tukey's honestly significant difference test at $P < 0.05$. The correlation analysis was carried out by the "Corrplot" package in R (version 0.93; https://github.com/taiyun/corrplot).

## RESULTS

### Induced-allelopathy of two rice cultivars treated with appropriate phytohormones under the optimum induction conditions

Preliminary screening results of 10 plant hormones with three concentrations were presented in Table S1. Compared with the other plant hormones, low concentrations of four exogenous plant hormones (ETH, ABA, BR and EBL) were selected more suitable for the growth induction of the roots of allelopathic rice PI and non-allelopathic rice LE. Further screening Experiment 1 and Experiment 2 were run to obtain the optimum induction phytohormones, evaluating the changes of the induced-allelopathy of PI and LE under induction treatment of ETH, ABA, BR and EBL at different concentrations. Compared with the inhibition rate of rice accessions without induction (defined as genetic allelopathy, GA), the inhibition rate of two rice accessions with induction of ETH, ABA, BR and EBL (defined as total allelopathy, TA) were increased with different concentrations in both PI and LE (Fig. S1). Based on the results of the induced-allelopathy (IA, IA = TA–GA) of four hormones between the two rice cultivars in Fig. S2, we will select three, five μmol/L ABA and 0.3, 0.5 nmol/L EBL for induction with different time gradients to analyze the time-effect of induced-allelopathy in the two rice cultivars.

As shown in Figs. 1A and 1B, regardless of ABA or EBL induction, the induced-allelopathy (IA) of non-allelopathic rice LE was greater than that of allelopathic rice. Although the peaks of IA of the two rice varieties after treatment with the same hormone appeared at different treatment times, it is more practical to improve the allelopathy of non-allelopathy rice. Therefore, we focused on the increment of IA in non-allelopathy rice. We considered that a treatment time of 3d to be the appropriate induction time for both rice cultivars, and the optimal plant hormones were three μmol/L ABA (Fig. 1A) and 0.5 nmol/L EBL (Fig. 1B). Under these induction conditions, the inhibitory impact of leaf extracts and exudates of roots from varieties of rice was markedly enhanced. After treatment with three μmol/L ABA, the total allelopathy (TA) values of rice root exudates were 39.82% in PI, and 32.17% in LE. The induced-allelopathy (IA) values (9.62%) accounted for 23.9% of TA values in PI, while IA values (13.76%) in LE accounted for 43.1% of TA values (Fig. 1C). As shown in Fig. 1D, treatment with 0.5 nmol/L EBL also substantially enhanced the inhibitory effect of the two rice cultivars. The TA values in PI and LE induced by 0.5 nmol/L EBL were 37.42% and 31.33%, respectively. The IA values induced by EBL were 7.83% in PI which accounted for 20.4% of TA values, whereas the IA values in LE were 11.51% that accounted for 36.7% of TA values (Fig. 1D). The trends in the IA and TA values of the two rice leaf extracts were in agreement with the results of their root exudates. Obviously, the extent of the increase in the induced-allelopathy (IA/TA) of non-allelopathic rice was markedly greater than that in allelopathic rice.

### Fine-root traits in allelopathic and non-allelopathic rice cultivars treated with two appropriate phytohormones

As shown in Figs. 2A–2D, ABA treatment increased the root length, root surface area and root volume in the 0–0.2 mm diameter range of PI root by 1.49, 1.85 and 1.40-fold, respectively. LE treated with ABA showed a maximal enhancement in root surface area

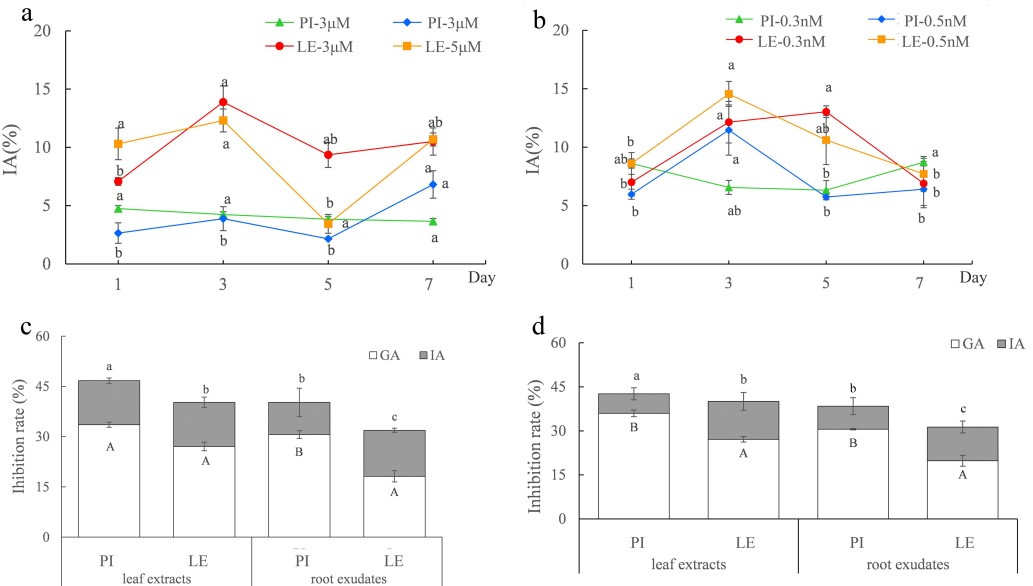

**Figure 1** **The allelopathic potential of two rice cultivars (PI-PI31277; LE-Lemont) treated with ABA or EBL under the optimum induction conditions.** (A–B) Effects of ABA (A) or EBL (B) on the induced-allelopathy (IA) of rice leaf extracts at different induction times. The lowercase letters denote significant variations at $p < 0.05$ levels among various treatment durations of the identical hormone concentration. (C–D) The inhibitory rate (%) of the leaf extracts and root exudates of two rice on lettuce root length treated with three μmol/L ABA (C) or 0.5 nmol/LEBL (D), for 3 days. GA, genetic allelopathy, untreated control; IA, induced-allelopathy. The lowercase letters denote substantial variations in total allelopathy at $p < 0.05$ levels between different rice leaf extracts and root exudates. The uppercase letters indicate significant differences of induced-allelopathy at $p < 0.05$ levels between various rice leaf extracts and root exudates.

within the 0.2–1.0 mm diameter range. After EBL treatment, the root length, root surface area, and root volume of PI increased exhibited the most substantial increases at a diameter range of 0–0.2 mm, which were 1.95 times, 2.49 times, and 1.85 times that of the control, respectively. The root length, root volume, and root surface area of LE increased most in the diameter range of 1.0–2.0 mm, which were 1.56, 1.73 and 1.57 times greater than those of the control, respectively, after EBL treatment. Regardless of ABA or EBL induction, The quantity of tips of root and the dry weight of roots in two rice cultivars exhibited a considerable increase (Figs. 2E–2F).

## Appropriate phytohormones induced the production of rice allelochemicals

The production of secondary metabolites by rice during the seedling stage is the main mechanism for inhibiting weed growth. Based on previous reports (*Li et al., 2020*), this study examined the impact of exogenous hormones on the levels of phenolic acids and terpenoids. Eight phenolic acids (protocatechuic, p-hydroxybenzoic, vanillic, syringic, p-coumaric, ferulic, salicylic, and cinnamic acid) are identified as allelochemicals in rice (*Li et al., 2019*). When the seedlings of the two rice cultivars were exposed to ABA or EBL under the optimum induction conditions, the contents of some phenolic acids increased in both

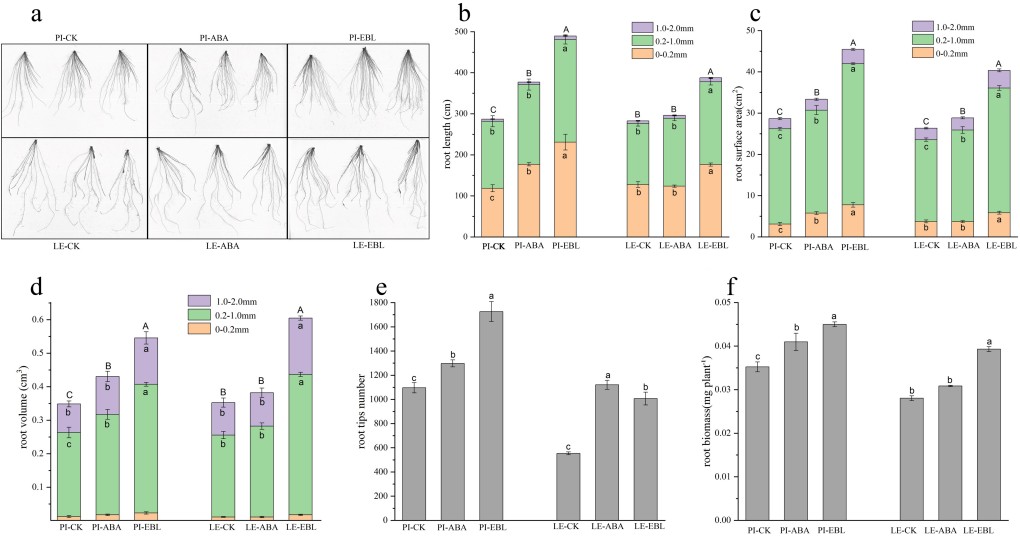

**Figure 2** **Fine-root traits of two rice cultivars treated with three μmol/L ABA or 0.5 nmol/L EBL for 3 days.** PI-CK, allelopathic cultivar PI312777 without treatment; PI-ABA, allelopathic rice PI312777 incubated with ABA; PI-EBL, allelopathic cultivars PI312777 incubated with EBL; LE-CK, non-allelopathic rice Lemont without treated; Le-ABA, non-allelopathic rice Lemont incubated with ABA; LE-EBL, non-allelopathic rice Lemont incubated with EBL. (A) Schematic diagram of rice root system scanning. (B–D) Aggregate value of root length (B), root surface area (C), and root volume (D) in the fine-root diameter range of 0–0.2, 0.2–1.0 and 1.0–2.0 mm of two rice cultivars. Significant differences ($P < 0.05$) between different treatments in the same fine-root diameter range of the same rice cultivars were indicated by different lowercases, and the significant differences for total values of 0–2.0 mm diameter were indicated by different uppercases at the end of the bars, according to Tukey's honestly test. (E–F) Root tips number (E) and root biomass (F) of two rice cultivars treated with ABA or EBL. The letters in lowercase indicate substantial variations at the $P < 0.05$ level across the different treatments of the identical rice cultivars.

rice leaf- extracts and root-exudates, while some decreased compared with the untreated control (Figs. 3A–3B). However, upon aggregating the concentrations of eight phenolic acids, it was observed that the cumulative levels of the selected phenolic allelochemicals in the hormone-treated groups were significantly higher in both PI and LE compared to the control group, regardless of the levels of total phenolic acids in rice leaves and roots (Figs. 3A–3B). In particular, the total contents of eight phenolic acids present in the root-exudates of PI and LE induced by EBL were 1.16 and 2.33 times greater, respectively , relative to the untreated control (Fig. 3B). At the same time, ABA treatment also elevated the concentration of phenolic acids secreted by PI and LE roots by 1.48 and 1.87 times, respectively.

GC−MS analysis results identified approximately 30 terpenoids in rice leaves, mainly including monoterpenoids, sesquiterpenoids and diterpenoids. The contents of each group were summarized and analyzed. As shown in Fig. 3C, the terpenoids in the two rice leaves of the control and treatment groups were mainly monoterpenes and sesquiterpenoids, and The levels of these two categories in the hormone-treated group were significantly higher compared to the control group, regardless of PI and LE. In addition, the contents of

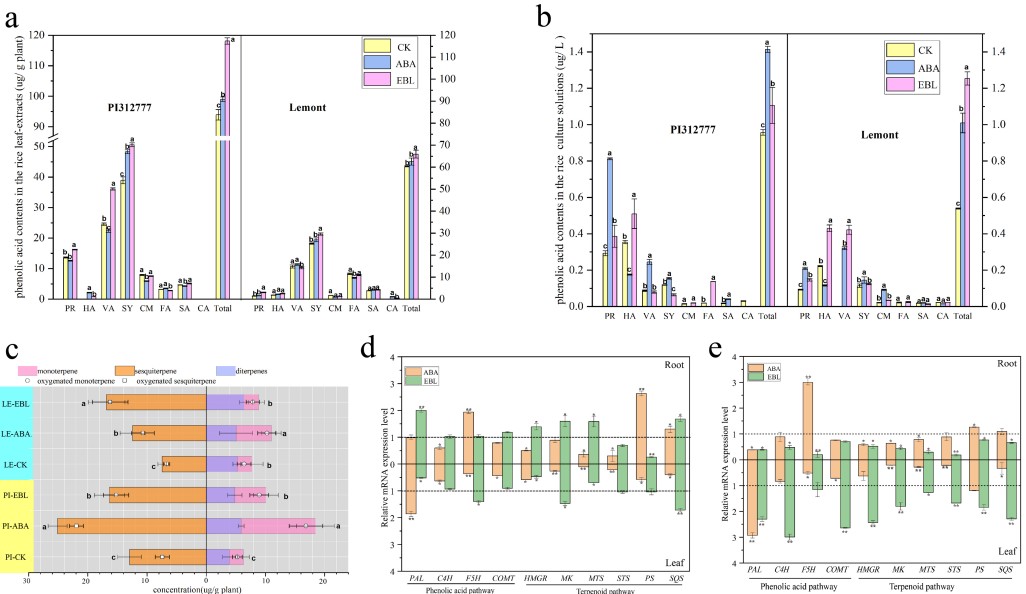

**Figure 3** **The contents and expression profiles of two important types of rice allelochemicals treated with three μmol/L ABA or 0.5 nmol/L EBL for 3 days.** PI-CK, allelopathic cultivar PI312777 without treatment; PI-ABA, allelopathic rice PI312777 incubated with ABA; PI-EBL, allelopathic cultivar PI312777 incubated with EBL; LE-CK, non-allelopathic rice Lemont without treated; LE-ABA, non-allelopathic rice Lemont incubated with ABA; Le-EBL, non-allelopathic rice Lemont incubated with EBL. (A–B) The contents of eight phenolic acids in leaf- extracts (A) and root-exudates (B) of two rice cultivars. PR, protocatechuic acid; HA, 4-hydroxybenzoic acid; VA, vanillic acid; SY, syringic acid; CM, p-coumaric acid; SA, salicylic acid; FA, ferulic acid; CA, cinnamic acid; Total, sum of eight phenolic acids. (C) The contents of different terpenoids in the leaf- extracts of two rice cultivars. (D–E) Relative expression levels of five key genes involved in the biosynthesis of phenolic acids (*PAL*, *C4H*, *F5H* and *COMT*) and terpenoids (*HMGR*, *MK*, *MTS*, *STS*, *PS* and *SQS*) in the leaves and roots of allelopathic cultivar PI312777 (D) and non-allelopathic rice Lemont (E) incubated with ABA or EBL. The gene expression levels are indicated in terms of the fold induction compared with the control incubated without ABA or EBL. *PAL*, phenylalanine ammonia-lyase; *C4H*, cinnamate-4-hydroxylase; *F5H*, ferulic acid-5-hydroxylases; *COMT*, caffeic acid 3-O-methyltransferase; *HMGR*, 3-hydroxy-3-methylglutaryl-CoA reductase; *MK*, mevalonate kinase; *STS*, sesquiterpene synthase; *PS*, phytoene synthase; *SQS*, squalene synthase. The various letters represent the significant differences between the control and treatments within the same rice ($P < 0.05$).

oxygenated terpenoids in the rice leaves were further compared. Two hormones increased the ratio of oxygenated terpenoids, especially the ratio of the contents of oxygenated sesquiterpenes in total sesquiterpenes. ABA treatment increased the ratio of the contents of oxygenated sesquiterpenes in total sesquiterpenes from 7.41% to 21.92% in PI, and from 6.67% to 10.7% in LE; After EBL induction, the ratio of oxygen-containing sesquiterpenes in PI was increased to 15.23%, while the ratio in LE was increased to 16.24%.

## Expression patterns of key genes involved in the biosynthesis of rice allelochemicals

When two rice cultivars were incubated with ABA or EBL, ten genes involved in the biosynthesis of phenolic acids (*PAL*, *C4H*, *F5H* and *COMT*) and terpenoids (*HMGR*, *MK*, *MTS*, *STS*, *PS* and *SQS*) were expressed in variety-dependent manners (Figs. 3D–3E).

For four genes from the phenolic acid pathway, ABA treatment significantly induced the expression of *F5H* in the roots and *PAL* in the leaves of both PI and LE, while expression of phenolic acid biosynthesis genes *PAL*, *C4H*, and *COMT* in rice leaves of LE was significantly up-regulated by EBL, and *PAL* induction also occurred in rice roots of PI. In particular, the expression of these four genes was much stronger in both tissues of non-allelopathic rice LE than allelopathic rice PI, regardless of ABA or EBL application. Furthermore, ABA and EBL also regulated gene expression levels of terpenoids biosynthesis genes in rice tissues of both PI and LE. EBL strongly induced (more than twofold changes) *HMGR* and *SQS* in LE leaves, the expression of *MK* and *SQS* in both tissues of PI were also increased after EBL treatment. However, ABA decreased the all of the biosynthesis genes in the leaves of both the allelopathic and the non-allelopathic variety. Compared with the untreated control group, the expression of *PS* in rice roots of PI was strongly increased (more than twofold changes) and were less strongly induced in LE roots.

## Correlation between root morphological traits and induced-allelopathic potential in rice after hormone induction

According to the experimental results in Fig. 2, root length and root volume with diameters ranging from 0–0.2 mm and 0.2–1.0 mm were selected to represent root morphological indexes. At the same time, the induced-allelopathy (IA) of rice root exudates (Figs. 1B and 3C), the contents of allelochemicals (phenolic acids and terpenoids) (Figs. 3B and 3C) and the expression level of genes related to allelochemical synthesis (Figs. 3D and 3E) represented the allelopathy potential of rice, and the correlation between two hormones' induced allelopathic potential and root morphological features, was investigated.

The correlation results showed that the induced-allelopathy (IA) after EBL application (Fig. 4A) was significantly positively correlated with the root volume with a diameter of 0.2–1.0 mm, phenolic acid contents in the root-exudates, the expression levels of phenolic acid synthesis genes (*C4H*,*COMT*) and terpenoid synthesis genes (*MK*, *STS*, *PS* and *SQS*). Under ABA treatment, IA was also positively correlated with root length with a diameter of 0.2–1.0 mm, phenolic acid contents in the root-exudates and terpenoid contents in rice leaves (Fig. 4B). However, IA was significantly negatively correlated with the expression levels of the phenolic acid synthesis genes (*C4H*, *F5H*, *COMT*), the terpenoid synthesis genes (*MK*, *MTS*, *PS*, *STS*) ($P < 0.01$), and the *MK* and expression levels ($P < 0.05$).

## DISCUSSION

Research has shown that introducing external chemical substances can enhance the allelopathic properties of rice, which can be utilized as a method for controlling weeds (*Bi et al., 2007*; *Fang et al., 2009*; *Kong et al., 2004a*; *Xu et al., 2010*; *Zhang et al., 2019*). It has also been suggested that most native rice cultivars possess high allelopathic potential because the genes against weeds in allelopathic accessions are also present in all native rice accessions (*Courtois & Olofsdotter, 1998*). However, this trait has been weakened or lost during hybridization and artificial selection for other traits, such as high yield or resistance to diseases and pests (*Rahaman et al., 2021*; *Gealy, Rohila & Botkin, 2019*). In this study, the genetic allelopathy (GA) of allelopathic rice PI was greater than that of

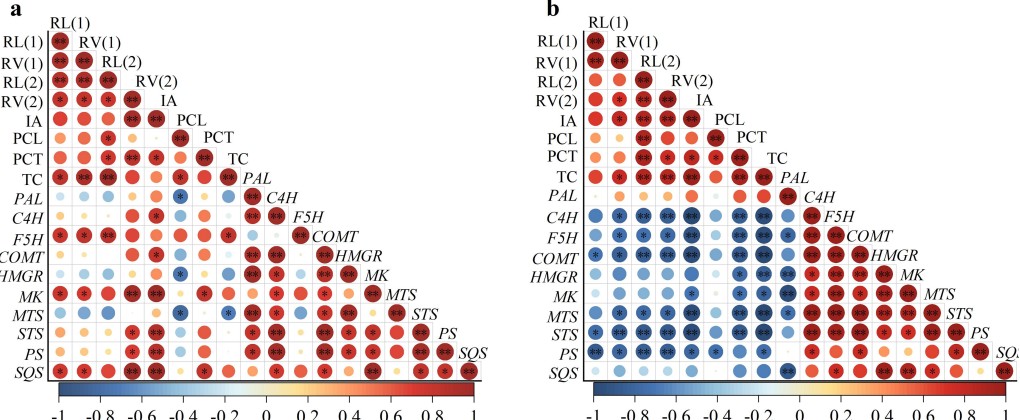

**Figure 4** **Correlation between root morphology and induced-allelopathic potential in rice after EBL (A) and ABA (B) treatment.** RL(1), root length at 0–0. 2 mm; RL(2), root length at 0.2–1.0 mm; RV1, root volume at 0–0.2 mm; RV2, root volume at 0.2–1.0 mm; IA, induced-allelopathic potential; PCL, total phenolic acid contents of rice leaf-extracts; PCT, total phenolic acids contents in root-exudates; TC, total terpenoid contents in leaves in rice leaf-extracts. Circle size and color shade represent correlation coefficients, * and ** represent significant differences at the $P < 0.05$ and $p < 0.01$ levels, respectively.

non-allelopathic rice, so the total allelopathy (TA) of allelopathic rice after phytohormones induction greater. However, the extent of increments in induced-allelopathy (IA/TA, IA = TA-GA) of non-allelopathic rice was significantly greater than that in allelopathic rice PI (Figs. 1C and 1D), which was consistent with the conclusions obtained by *Zhang et al. (2018)* and *Zhang et al. (2019)* using exogenous root-exudates of barnyardgrass to increases rice allelopathy. Furthermore, our findings indicate that he total contents of phenolic acids, which have been recognized as important allelochemicals in rice, were significantly increased in the leaf extracts and root exudates of both PI and LE in comparison to the control group (Figs. 3A and 3B). Although the increase of total phenolic acids contents in leaf-extracts of non-allelopathic rice LE was lower than that of allelopathic rice PI (Fig. 3A), the increment of total phenolic acids contents in root-exudates of LE induced by ABA or EBL was significant, the increase of total phenolic acids after EBL induction was especially much greater than that of allelopathic rice. This may explain why IA (induced-allelopathy) is more effective in non-allelopathic rice LE. In actual cropland ecosystems, most rice varieties have low or no allelopathic potential. This finding reminds us that enhancing induced-allelopathy of conventional rice varieties at its seedling stage through chemical induction may be a more practical approach for applying rice allelopathy in rice production.

Our hypothesis proposed the use of exogenous plant hormones to improve weed-resistance by regulating rice roots at the seedling stage, which was proven to be true for two rounds of screening experiments in which root morphology at seedling stage was used as a screening target in hydroponic experiments (Table S1). Exogenous ABA mainly affected the root length and root volume of rice fine-roots in the 0–0.2 mm diameter range at the seedling stage, while the induction by EBL mainly affected the root morphology in the 0–0.2 mm and 1.0–2.0 mm diameter ranges (Figs. 1B and 1D).

Research has demonstrated that allelopathic rice inhibits the growth of surrounding weeds by releasing allelochemicals into the environment *via* its root system (*Kong et al., 2019*; *Wang et al., 2021*) which was significantly positively correlated with the fine-root traits of allelopathic rice at the seedling stage (*Li et al., 2019*). Correlation analysis confirmed that fine-root morphology (root length and root volume with a diameter of 0.2–1.0 mm) was significantly positively correlated with the total content of phenolic acid and terpenoids and with the induced-allelopathic activity (Fig. 4). These results open up a new avenue for modifying the root structure of rice to enhance its allelopathic effects through chemical induction by screening for phytohormone regulators. It must be pointed out that the spatial distribution of allelopathic rice roots in paddy soil appears to have important impact on its weed-suppressive activity at the seedling stage (*Li et al., 2022*), and more in-depth work is needed to explore the inductive effect of plant hormones under field conditions.

In most cases, allelopathic interference is the result of the concerted action of different secondary metabolites (*Kostina-Bednarz, Płonka & Barchanska, 2023*). Earlier transcriptomic studies have indicated that various pathways, including "phenylpropanoid biosynthesis", might play an essential role in controlling rice and barnyardgrass interactions (*Zhang et al., 2019*; *Sultana et al., 2023*). The initial stages in the synthesis of compounds derived from phenylpropanoids are facilitated by the enzyme phenylalanine ammonia lyase (*PAL*), cinnamate 4-hydroxylase (*C4H*), and p-coumaroyl coenzyme A ligase (*4CL*). In this context, PAL is integral to the phenylpropanoid pathway, facilitating the biosynthesis of a wide variety of phenolic compounds., including flavonoids, lignin, and other phenylpropanoid derivatives. Earlier transcriptomic studies have also highlighted the positive regulatory role of PAL in rice's allelopathic potential (*Fang et al., 2016*; *Fang et al., 2020*; *Zhang et al., 2019*). *García-Romeral et al. (2024)* investigated the genetic basis of allelopathy in japonica rice grown in temperate regions through a genome-wide association study (GWAS) and again confirmed four genes coding for PAL could be potentially related to rice allelopathy. The data produced in the course of this research also revealed that the expression of *PAL* was up-regulated in rice tissues of PI and LE by ABA or EBL treatment (Figs. 3D, 3E). In addition, EBL also significantly up-regulated the expression levels of *C4H* and *COMT* genes in the leaves of non-allelopathic rice LE, but the increasing degree of the contents of total phenolic acids in LE leaves induced by EBL was weaker than that of PI (Fig. 3A). This inconsistent trend may be due to the fact that up-regulation of *C4H* gene expression in LE rice leaves may lead to the synthesis of other phenolic compounds such as flavonoids or lignins. Allelopathic rice varieties Huagan-3 can release flavonoid allelochemicals tricin (5, 7, 4′-trihydroxy-3′, 5′-dimethoxyflavone) (*Kong et al., 2004b*; *Kong et al., 2019*). (−)-Loliolide is reported as a signaling chemical that upregulates the expression of two key genes (*CYP75B3* and *CYP75B4*) involved in the biosynthesis of flavonoid tricin and the production of tricin in allelopathic rice roots (*Li, Zhao & Kong, 2020*). In further studies, it will be identified the effect of exogenous EBL treatment on the production of tricin or other flavonoid and the expression level of their biosynthetic candidate genes.

Allelopathic interaction involves a complex interplay among plants rather than a straightforward, singular pathway (*Sultana et al., 2023*). *Zhang et al. (2019)* also indicated

six key genes (*HMGR*, *MK*, *MTS*, *STS*, *PS* and *SQS*) were involved in the biosynthesis of terpenoids. In the mevalonate (MVA) pathway, hydroxymethylglutaryl-CoA reductase (HMGR) converts hydroxymethylglutaryl-CoA to mevalonate, which is considered to be one of the most important enzymes regulating the rate-limiting step in the MVA metabolic pathway. Subsequently, mevalonate kinase (MK) catalyzes a phosphorylation reaction that converts MVA to isoprene phosphate (IPP), which are precursors for all downstream products such as monoterpenes, sesquiterpenes, and diterpenes (*Buchanan, Gruissem & Jones, 2000*). Current research indicates that most of these genes (*HMGR*, *MK*, *SQS*) can be induced by EBL (Figs. 3D, 3E), which was consistent with the results of *Zhang et al. (2019)*. Combining the expression level of the six genes with the total contents of terpenoids in rice leaves, the findings showed a steady pattern (Fig. 3C). However, this consistent tendency was absent under ABA treatment: the contents of total terpenoids induced by ABA was increased, but all of the genes' relative expression levels in rice leaves were down-regulated, except *PS* in the roots of both PI and LE (Figs. 3D, 3E). *Yin et al. (2020)* also reported that the expression of key triterpene synthesis genes *HMGR*, *BPW*, *SE* and *BPY* in birch could be down-regulated by exogenous application of ABA.In addition, ABA and EBL significantly increased the ratio of oxygenated terpenoids, which were not further specified due to the limitations of semi-quantitative analysis by GC-MS. Terpenoid chemicals produced in rice leaves and variations in gene expression could be connected to the creation of additional downstream metabolites in the MVA metabolic pathway. Many studies have confirmed that the oxygenated diterpene momilactone B is considered another potent rice allelochemical (*Kato-Noguchi, 2011*), and the products of seven genes (*CPS4*, *KSL4*, *CYP99A3*, *MAS*, *CYP701A8* , *CYP76M14* and *CYP76M8*) on chromosomes 1, 2, 4, and 6 synthesize momilactone B (*Xu et al., 2012*; *Serra, Shanmuganathan & Becker, 2021*). Further studies will explore whether exogenous ABA treatment can directly affect the synthesis of momilactone B and the expression level of its potential biosynthesis genes.

In this study, the induced-allelopathy of rice after EBL induction was positively correlated with the expression levels of *C4H*, *COMT*, *MK* and *SQS* (Fig. 4A), while that was negatively correlated with the expression levels of *C4H*, *F5H*, *COMT*, *MTS* and *STS* after ABA treatment (Fig. 4B). There is still a lack of direct data on the molecular mechanisms underlying the induction of rice allelopathy by ABA or EBL. *Fang et al. (2020)* suggested that a MYB transcription factor, *OsMYB57* regulates MAPK11 to interact with PAL2;3 and modulate rice allelopathy. The high activity of *OsPAL2;3* in the allelopathic rice PI312777 was considered to contribute mainly to the increased synthesis of phenolic acids in rice plant. Research is ongoing to focus on whether hormonal induction has an effect on this transcription factor that regulates phenolic acid synthesis, which will further help to effectively improve the allelopathic potential of conventional rice and reduce herbicide use in rice production.

## CONCLUSIONS

The results of this study indicate that exogenous application of ABA or EBL can effectively enhance the induced-allelopathic in rice, particularly in non-allelopathic rice varieties.

The exogenous application of both hormones can increase the yield of allelochemicals in rice. However, there are differences in the ability of ABA and EBL to regulate the expression of genes involved in the biosynthetic pathways of these allelochemicals, suggesting that these two hormones may act through different molecular mechanisms. Overall, these results highlight the potential of ABA and EBL as tools for enhancing allelopathy in rice, offering promising avenues for sustainable weed management and improved agricultural productivity. Correlation does not imply causation, further research is needed to better understand the specific molecular pathways involved and to optimize the use of phytohormones in crop management strategies.

## ACKNOWLEDGEMENTS

We thank Prof. Xing Liu, and Prof. Changxun Fang for their valuable suggestions on the study design and careful corrections on the revision.

### Funding

Funding was provided by the Fujian Provincial Natural Science Foundation of China (grant no. 2024J01398) and the Science Innovative Program of Fujian Agriculture and Forestry University (grant no. KFb22055XA). The funders had no role in study design, data collection and analysis, decision to publish, or preparation of the manuscript.

### Grant Disclosures

The following grant information was disclosed by the authors:
Fujian Provincial Natural Science Foundation of China: 2024J01398.
Science Innovative Program of Fujian Agriculture and Forestry University: KFb22055XA.

### Competing Interests

The authors declare there are no competing interests.

### Author Contributions

- Ting Wang conceived and designed the experiments, performed the experiments, analyzed the data, authored or reviewed drafts of the article, and approved the final draft.
- Xinyi Ye conceived and designed the experiments, performed the experiments, analyzed the data, authored or reviewed drafts of the article, and approved the final draft.
- Yuhui Fan analyzed the data, prepared figures and/or tables, and approved the final draft.
- Shuyu Chen analyzed the data, prepared figures and/or tables, and approved the final draft.
- Huayan Ma analyzed the data, prepared figures and/or tables, and approved the final draft.
- Jiayu Li conceived and designed the experiments, performed the experiments, authored or reviewed drafts of the article, and approved the final draft.

## Data Availability

The raw data is located in the Supplemental Information.

## Supplemental Information

Supplemental information for this article can be found online at http://dx.doi.org/10.7717/peerj.19700#supplemental-information.

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
