# Peer review of "Study on the induction of exogenous plant hormones to enhance the weed suppression ability of allelopathic and non-allelopathic rice accessions"

_PeerJ, doi:10.7717/peerj.19700_

## Round 0.1 · original submission · Minor Revisions

Allelopathic impacts of the rice accessions on weeds are significant in preventing yield and quality losses. Although your work contains valuable information about the practical application of allelopathy, it does not fully meet some requirements of our journal. Therefore, it is essential to address certain technical details to enhance the article further. I strongly recommend carefully reviewing the reviewers' suggestions and thoughtfully considering each recommendation. If you disagree with any suggestion, it would be helpful to provide clear, well-reasoned justifications for your viewpoint.

Reviewer 1 ·

Basic reporting

The manuscript is written in technical English; however, punctuation and spelling errors also need correction (e.g., Lines 78, 79, 97, 133, 296, 346, 354, 368, 370). The term 'Pheynlalanine' throughout the manuscript should be corrected to 'Phenylalanine'.
The introduction offers a satisfactory overview of allelopathy in rice and cites relevant literature. However, the rationale for enhancing induced allelopathy (IA) via phytohormone application should be clarified and strengthened, particularly in Lines 57–86. The authors can elaborate on the practical implications of promoting IA over genetically determined allelopathy (GA), especially in non-allelopathic rice varieties.
The manuscript adheres to the IMRaD structure and includes all essential sections. The Methods section is logically organised with clear subsections. Figures are generally well-designed, but figure captions need revision to explicitly describe the content of each panel (e.g., Figure 1 panels a–f).
The manuscript refers to supplementary tables, and raw data have been deposited in an open-access repository, which supports transparency and reproducibility.

Experimental design

The study addresses a relevant and novel research gap—enhancing allelopathy in non-allelopathic rice through exogenous phytohormone application. The research question is clearly stated and logically developed. Experiments were conducted under controlled hydroponic conditions with appropriate replication and randomisation, adhering to acceptable scientific standards. The methodology is detailed, including specific hormone concentrations, sampling times, analytical techniques (HPLC, GC-MS, qPCR), and software tools (SPSS, Corrplot in R). This level of detail facilitates reproducibility. Minor suggestion: The authors can state the statistical assumptions tested (e.g., normality, homoscedasticity) prior to performing ANOVA.

Validity of the findings

Results are clearly presented in both figure and tabular formats, with underlying data systematically collected and made accessible. This enhances transparency.
The statistical approach is generally appropriate. One-way ANOVA followed by Tukey’s HSD was applied to compare treatment effects. Correlation matrices generated with the Corrplot R package were used to validate interrelationships among traits. The definition of error bars (e.g., standard error vs. standard deviation) is unclear in several figures and should be clarified. Although statistical significance is reported in the study, the lack of effect size measures (e.g., eta-squared or partial eta-squared) and statistical power makes it difficult to assess the biological relevance of the findings. Effect size indicates the proportion of variance explained by a given factor, and thus it is recommended to interpret the results not only based on p-values but also in conjunction with these complementary metrics. The authors should specify what the error bars represent in all figures. Additionally, reporting effect size metrics would help readers better understand the practical relevance of the observed effects, beyond p-values.
The study integrates morphological, biochemical, and molecular data in a coherent manner: Hormonal treatments (ABA and EBL) altered specific root traits by diameter class.These traits correlated with phenolic acid and terpenoid concentrations. Upregulation of biosynthetic genes (PAL, C4H, SQS) under EBL treatment aligns with metabolite profiles. This integrative approach enhances internal validity. However, a discrepancy is observed under ABA treatment: increased metabolite accumulation occurs without corresponding gene expression changes.
The observed mismatch between qPCR data and metabolite levels under ABA treatment is biologically plausible. The authors should explore these possible mechanisms in the Discussion section and avoid deterministic interpretations of gene–function relationships.
While the study presents strong correlations between morphological traits, metabolite levels, and allelopathic effects, these should not be interpreted as causative. The authors should clearly state that correlation does not imply causation and recommend further validation (e.g., mutant analysis, enzyme assays) for establishing causal links.
Experimental methods are described in sufficient detail for replication. Consistent results across multiple screening rounds strengthen reliability. Use of established tools and instrumentation (e.g., SPSS, GC-MS, qPCR) further supports robustness. The findings are supported by multi-level evidence, statistically sound methods, and coherent biological reasoning. Improving reporting effect sizes, clarifying statistical assumptions, and appropriately interpreting molecular data will enhance the manuscript’s scientific rigour.

Additional comments

The study presents several strengths, including comprehensive data encompassing morphology, biochemistry, and gene expression. It effectively highlights the relationship between root traits and allelopathic potential, and it demonstrates the enhancement of allelopathy in rice accessions that were previously non-allelopathic. These findings have promising agronomic implications for sustainable weed management.This is a well-executed and innovative study with practical implications. It is recommended that the manuscript focus particularly on statistical elaboration and detailed figure legends.

Reviewer 2 ·

Basic reporting

The ms "Increasing the weed-suppressive activity of allelopathic and non-allelopathic rice accessions by induction treatment with exogenous abscisic acid or 24-epibrassinolide" showed the effect of plant hormones on the induction of allelopathy in two rice accessions (allelopathic PI312777 and Lemont). The authors measured secondary metabolites (i.e., phenols and terpenoids) on rice-treated seedlings, assayed the allelopathic effect on lettuce, and evaluated the root morphology of the two accessions of rice. The authors found interesting results that contribute to weed management by improving rice's allelopathic properties.
Next, I state some comments that could be useful for the authors:
Comment 1 (C1): Although the manuscript is clear and the language is understandable, I suggest the author go deep in explaining the action mechanisms of the phytohormones they chose. It is not enough to say L 114-118 "There is a wealth of information that some classical plant hormones such as abscisic acid, gibberellin, and indoleacetic acid, etc., or new hormones such as salicylic acid, jasmonate, methyl jasmonate, brassinolide, and strigolactones, etc. can regulate the growth and development of plant roots individually or interactively (Zhu et al., 2005; Chen et al., 2006; Haider et al., 2018; Zhou et 118 al., 2019). "
For example, Methyl Jasmonate is a molecule related to herbivory that could act as an infochemical to receiver plants. I understand that this research aims to find some phytohormones that elicit the allelopathic response in rice plants (PI312777 and Lemont). Still, explain why you chose the mentioned phytohormones.
C2: Change the title of the manuscript because it is too specific, and the hypothesis and objective are broad. An alternative is to change experiment 1, including results, to Supplementary Material and keep exp 2 and exp 3 in the main text.
C3: Some details that should be reviewed:
Move lines 120-125 "In this study, we investigated the induction of allelopathic properties in common exogenous hormones in rice by using root morphology at seedling stage as a screening target in hydroponic experiments. We evaluated the changes of induced-allelopathy of rice with different allelopathic potentials under the exogenous application of different plant hormones. In addition, we further explored the underlying inductive mechanism by analyzing root morphology, allelochemicals production, and the expression levels of allelochemical synthesis related genes" and placed after Lines 110-113.
Line 57. Change "methods"
Line 61. Insert a space: "2009).The"
Line 68. Delete a point "rice tissues.."
C4: The Introduction is too extensive; please synthesize this section. For example, lines 95-97 could be integrated with the second part of the same idea (Lines 97-102). On the contrary, as you study the root system and mention (Lines 94-95), "The allelopathy between rice and weeds relies on the involvement of the root system in the 95 underground part of the plant." I suggest writing a pair of lines about how to differentiate the effect of chemical interference from resource competition.

Experimental design

The experimental design is adequate and detailed. The number of experimental units allows for reliable results. Please consider moving experiment 1 to supplementary material.
It is absolutely necessary to state clearly the values of F and the P-value with respect to statistical results.

Validity of the findings

No comments

Additional comments

The results of this research Will contribute to finding alternatives to herbicide use. Still, my significant observation is that the authors should be argumentative about selecting the phytohormones they used.

---

## Round 0.2 · accepted · Accept

I would like to thank you for accepting the referees' suggestions and improving your article based on their suggestions. Your article is ready to publish. We look forward to your next article.